# Altered Inflammatory State and Mitochondrial Function Identified by Transcriptomics in Paediatric Congenital Heart Patients Prior to Surgical Repair

**DOI:** 10.3390/ijms25137487

**Published:** 2024-07-08

**Authors:** Francesca Bartoli-Leonard, Amy G. Harris, Kelly Saunders, Julie Madden, Carrie Cherrington, Karen Sheehan, Mai Baquedano, Giulia Parolari, Andrew Bamber, Massimo Caputo

**Affiliations:** 1Bristol Medical School, Faculty of Health Sciences, University of Bristol, Bristol BS8 1UD, UK; a.harris@bristol.ac.uk (A.G.H.); m.baquedano@bristol.ac.uk (M.B.); m.caputo@bristol.ac.uk (M.C.); 2Bristol Heart Institute, University Hospital Bristol and Weston NHS Foundation Trust, Bristol BS2 8ED, UK; kelly.saunders@uhbw.nhs.uk (K.S.); julie.madden@uhbw.nhs.uk (J.M.); carrie.cherrington@uhbw.nhs.uk (C.C.); karen.sheehan@uhbw.nhs.uk (K.S.); g.parolari@uhbw.nhs.uk (G.P.); 3North Bristol NHS Trust, Westbury on Trym, Bristol BS10 5NB, UK

**Keywords:** congenital heart disease, RNAseq, right ventricle, immune cell

## Abstract

Congenital heart disease (CHD) remains the most common birth defect, with surgical intervention required in complex cases. Right ventricle (RV) function is known to be a major predictor of sustained cardiac health in these patients; thus, by elucidating the divergent profiles between CHD and the control through tissue analysis, this study aims to identify new avenues of investigation into the mechanisms surrounding reduced RV function. Transcriptomic profiling, in-silico deconvolution and functional network analysis were conducted on RV biopsies, identifying an increase in the mitochondrial dysfunction genes *RPPH1* and *RMPR* (padj = 4.67 × 10^−132^, 2.23 × 10^−107^), the cytotoxic T-cell markers *CD8a*, *LAGE3* and *CD49a* (*p* = 0.0006, *p* < 0.0001, and *p* = 0.0118) and proinflammatory *caspase-1* (*p* = 0.0055) in CHD. Gene-set enrichment identified mitochondrial dysfunctional pathways, predominately changes within oxidative phosphorylation processes. The negative regulation of mitochondrial functions and metabolism was identified in the network analysis, with dysregulation of the mitochondrial complex formation. A histological analysis confirmed an increase in cellular bodies in the CHD RV tissue and positive staining for both CD45 and CD8, which was absent in the control. The deconvolution of bulk RNAseq data suggests a reduction in CD4+ T cells (*p* = 0.0067) and an increase in CD8+ T cells (*p* = 0.0223). The network analysis identified positive regulation of the immune system and cytokine signalling clusters in the inflammation functional network, as there were lymphocyte activation and leukocyte differentiation. Utilising RV tissue from paediatric patients undergoing CHD cardiac surgery, this study identifies dysfunctional mitochondrial pathways and an increase in inflammatory T-cell presence prior to reparative surgery.

## 1. Introduction

Congenital heart disease (CHD) remains the most common birth defect, with surgical intervention required every 5 years in children [1] to every 15 years in adults [2], conferring a significant risk of mortality with each reintervention. In contrast to adult heart disease, which disproportionately affects the left ventricle, complex CHD can be characterised by increased right ventricular (RV) pressure, leading to RV hypertrophy and eventually failure, with RV function known to be a major predictor of survival [3,4]. 

Current treatment strategies for left ventricular dysfunction, such as β-blockers, iACE and Angiotensin II receptor blockers, fail to impact RV dysfunction in CHD patients. This lack of viable treatment is further compounded by the currently available diagnostic modalities failing to predict early-stage subclinical changes that occur in the RV, highlighting the knowledge gap in this pathology. Alongside RV dysfunction in CHD, low-level inflammation in the myocardial tissue in malformed hearts promotes further dysfunction over time [5]. Small animal studies have demonstrated that non-ischaemia heart failure and RV failure are orchestrated by the adaptive immune response [6,7], with an exhausted T-cell population identified in RV dysfunction at a single-cell level [8]. Understanding the inherent differences in RV tissue from CHD patients compared to controls may suggest new ways to adapt surgical treatments to improve outcomes in patients and highlight the critical yet subtle changes to pay attention to in post-surgical patients. Differentially expressed genes and associated pathways demonstrate the inherent differences between tissue obtained from the left and right side of the heart [4,5]; however, there is limited knowledge of how this compares to non-CHD tissue, mainly due to the difficulty in obtaining such tissue. Inflammatory pathways in the right ventricle are suggested to develop in utero [9]; however, the functionality of these pathways prior to reparative surgery is unknown. Notably, changes in the mitochondria-associated gene expression and subsequent energy production through oxidative phosphorylation can severely limit RV function, leading to pulmonary hypertension and heart failure [10].

Thus, this study hypothesised that activated inflammatory pathways and mitochondrial dysfunction may be present in the RV prior to and during surgical intervention, leading to an increase in inflammatory cell prevalence in the RV. An improved understanding of the complex mechanisms occurring in the RV in CHD patients may aid the development of therapeutics to mitigate mitochondrial dysfunction and improve RV outcomes post-surgery in this high-risk paediatric cohort. 

## 2. Results

Transcriptomic analysis was conducted on 14 patients and 4 controls to examine the transcriptomic profile of biopsies taken from RVs during cardiothoracic surgery (Appendix A), and the molecular landscape was assessed (Figure 1A). The principal component analysis of RV transcriptomic profiles demonstrated clear clustering based on the disease state with no cut-off applied (Figure 1B). The distance matrix of log-normalised transcript counts presented as a hierarchical heatmap clustered control and CHD tissue separately (Figure 1C), with no filtering applied. 

Following DESeq2 [11] analysis, the hierarchical clustering demonstrates the differences between control and CHD tissue (false discovery rate < 0.05 and abs[log fold-change] > 2), with little variation seen in the global transcriptomic profile between CHD patients (Figure 1D, Appendix A). Library normalisation demonstrated similar read numbers between disease and control tissue (Figure 2A), with accurate estimations of disperation (Figure 2B) and a clear relationship between the mean expression level and the log fold change of the genes in the DESeq2 analysis (Figure 2C). The mitochondrial dysfunction genes *RPPH1* (log2FC = 11.86 and padj = 4.67 × 10^−132^) and *RMPR* (log2FC = 8.80 and padj = 2.23 × 10^−107^) were significantly increased in CHD compared to the control, as was the proinflammatory mediator *ABL1* (log2FC = 1.79 and padj = 2.60 × 10^−39^). The mitochondrial regulator *CHCHD2* (log2FC = 3.89 and padj = 8.38 × 10^−117^) was upregulated in the control conditions alongside the ribosomal subunits *PSMA2* (log2FC = 3.39 and padj = 7.07 × 10^−166^) and *PRL31* (log2FC = 2.96 and padj = 3.10 × 10^−133^) (Figure 2D). The expression of hallmark CHD genes was assessed in RVs compared to the control, with the CHD RV samples showing a reduction in *TBX5*, JAG1, *NOTCH2* and *NOTCH1* (padj = 2.24 × 10^−7^, 1.38 × 10^−27^, 7.76 × 10^−31^ and 9.11 × 10^−7^, respectively). Notably, no change was detected in the *GATA4* or *GATA6* genes directly associated with CHD development [12] (padj = 0.79 and 0.80, respectively); however, *NKX2-5*, linked to 10 CHD cyanotic abnormalities, was increased in CHD compared to the control, highlighting the complex genetic variation in these patients (padj = 4.97 × 10^−16^) (Figure 2E). Following previous studies of dysregulated immune responses in CHD patients, the key adaptive immune cell markers were assessed in the RV T-cell helper marker *CD4* was significantly downregulated (*p* < 0.0001), as was the T-cell naivety marker *SELL* (*p* = 0.003). The cytotoxic T-cell marker *CD8a* was significantly increased in CHD compared to the control (*p* = 0.0006), as was the proinflammatory marker *caspase-1* (*p* = 0.0055). The CD8+ T-cell accumulation marker *LAGE3* and T-cell tissue localisation marker *CD49a* were both increased in CHD compared to the control (*p* < 0.0001 and *p* < 0.0118, respectively) (Figure 2F).

Utilising findings from single-cell datasets [8,13,14], the gene expression of hallmark cell marker genes was assessed (Figure 3A), demonstrating unique transcriptomic profiles between disease and control tissue. The histological analysis of RV biopsies obtained from the same patients with CHD (n = 5) and novel controls (n = 5) obtained from the Bristol Pathology Department (unique from the RNAseq samples) showed an increase in cellular bodies in CHD RV tissue compared to the control. The breakdown of type I collagen in line with increased RV fibrosis was observed in CHD compared to the control demonstrated by a lack of red/yellow fibres shown in the Picro-Sirus red staining (PSR). Immuno-histological staining of the immune markers CD45 and CD8 was positive in CHD and less stained in the control tissue (Figure 3B). The deconvolution of the transcriptomic profiles was conducted to assess the cellular composition of the RV biopsies, utilising a single-cell backbone of RV biopsies in dilated cardiomyopathy and healthy controls taken from adult donors [8] to predict cellular proportions. As expected, transcripts clustered as B cells had a unique transcriptome compared to other cell types, with cardiomyocytes and fibroblasts clustered more closely. T cells, NK cells, myeloid cells and endothelial cells clustered similarly (Figure 4A). Normalised proportions of each cell type (Figure 4B) demonstrated similarity between disease and control tissue, irrespective of specific CHD malformation. The estimated proportions of cell types in both control and CHD tissue (Figure 4C) predicted that cardiomyocytes made up the majority of the cells present in the RVs (48% ± 7.42 and 70% ± 16.41, respectively; *p* < 0.001). No difference was observed in the prevalence of fibroblasts (6.7% ± 1.51 and 6.2% ± 7.94; *p* = 0.4236) with the endothelial cell fraction exhibiting significantly more in the control compared to CHD (22% ± 3.63 and 9% ± 4.49; *p* = 0.0002). No difference in B cells was observed (1% ± 0.56 and 2% ± 0.90; *p* = 0.235), with a decrease observed in natural killer (3% ± 2.58 and 7% ± 2.11; *p* = 0.0043), myeloid (2% ± 1.07 and 5% ± 1.16; *p* = 0.0001) and CD4+ T cells (2% ± 1.90 and 6% ± 2.76; *p* = 0.0067) in CHD compared to the control. Notably, CD8+ T-cell prevalence was greater in the CHD population compared to the control (7% ± 4.7 and 4% ± 2.9; *p* = 0.0223) (Figure 4D), with no association between CHD pathology and CD8+ prevalence. 

Transcriptomic analysis identified 10440 gene transcripts in the control and 9795 in CHD, with 82% shared homology (Figure 5A). The gene set enrichment analysis of the differentially expressed genes upregulated in the CHD RV biopsies identified the top 15 pathways from the Human Phenotype Ontology, KEGG, WikiPathways, Hallmark and Reactome databases, accessed through the fgsea v3.19 R package. Mitochondrial pathways were predominantly enriched in the CHD samples, with abnormalities in the mitochondria and changes in the mitochondrial respiratory electron transport chain (OXPHOS) reported (Figure 5B). The functional network analysis visualised non-redundant biological terms for large gene clusters in the transcriptomic profile of CHD RV biopsies (Figure 5C). Five main clusters were identified pertaining to mitochondrial functions, metabolism, inflammation, apoptosis and signalling pathways. Mitochondrial-related functional clusters were the most significantly upregulated in CHD, which were closely linked to dysregulated metabolic functions. Positive regulation of the immune system and cytokine signalling clusters was increased in the inflammation functional network, as well as lymphocyte activation and leukocyte differentiation, giving rationale to further assess the immune cell compartment and metabolic function in the right ventricles. 

## 3. Discussion

Surgical intervention for CHD is currently the only form of treatment; however, the majority of these patients require multiple reinterventions throughout their lives, with fibrosis, occlusion and calcification limiting the lifespan of implanted materials [15,16,17]. Animal models, predominantly murine and porcine, recapitulate the surgical aspect of CHD; however, due to the abrupt change from *healthy* to pathogenic through intervention, they fail to recapitulate the developmental aetiology of CHD and the slow changes in pressure and cellular composition. Thus, this study set out to profile tissue taken directly from patients immediately prior to surgery to better understand the differential cellular landscape in the right ventricle as a means to identify potential targetable differences between disease and control tissue. By transposing RNAseq data onto a single-cell backbone, cellular ratios can be estimated to elucidate inflammatory cells potentially active in RV tissue prior to intervention, which may impact the response of the localised tissue to implantable grafts, leading to premature failure and the need for repeated interventions. Landmark studies have highlighted the increased pro-inflammatory markers present in the myocardium of CHD patients [18,19], with an altered immune profile reported in patients who underwent thymectomies routinely as part of cardiothoracic surgery [20], exhibiting an increased level of T-cell expression. The data presented here demonstrate a CD8+ T-cell-rich RV environment, identified through CD8+ T-cell accumulation and the infiltration marker LAGE3, tissue residency marker CD49a+, pro-inflammatory cytokine caspase-1 RNA expression and positive CD8+ and CD45+ histological staining. This data, aligned with the network analysis predicting changes in leukocyte differentiation and lymphocyte activation, suggest a shift to a pro-inflammatory profile in the RV, which is not observed in control tissue. Previous studies have identified an increase in pro-inflammatory cells in adult heart-failure patients [8] and genetic malformations, such as DiGeorge syndrome [21], but have been thus far unconfirmed in non-genetically influenced CHD pathologies alone. 

Notably, the data shown here differ from previous findings in failing hearts, often conducted on the left ventricle [13,14], in that the cardiomyocyte numbers are increased in CHD tissue, and the myeloid cells are reduced. Heart failure is an acquired and metabolically active process [22], in which cardiomyocytes often undergo apoptosis and fibroblast transition to a myofibrotic phenotype, and with a different aetiology to congenital heart pathologies. In paediatric settings, however, cardiomyocyte apoptosis and cardiac necrosis are little understood, and the inherent differences between the right and left ventricles remain unclear [23], clearly requiring further investigation. Due to the difficult nature of accessing tissue biopsies prior to surgery to assess risk, the identification of circulating markers, such as increased inflammatory cell populations or their subsequent cytokine profile, may be able to act as surrogates to better understand the risk for these patients pre-surgery, though further targeted studies must be undertaken to fully elucidate this tentative relationship. In adult populations, the pre-operative inflammatory status has been demonstrated to impact surgical cardiac outcomes [24], with a systemic inflammatory index [25], currently utilised in cancers and stroke, able to accurately predict outcome favourability; however, the efficacy of using such a scoring system in a paediatric population remains unfounded. 

The network analysis conducted on the differentially expressed RNA profiles in CHD confirmed an inflammatory profile and identified a network of mitochondrial dysfunction-associated genes, suggesting a change in the RV metabolic profile prior to surgical intervention. Mitochondrial dysfunction defines T-cell exhaustion [26,27], and accumulating evidence suggests that mitochondrial dysfunction and low-grade inflammation play a critical role in the development of RV maladaptation [28], the progression of inflammation and accumulation of inflammatory cells in the regions of dysfunction [29,30]. In the RV CHD analysis, increased dysfunction in the oxidative phosphorylation and the electron transport chain were identified in relation to the increased expression of *CHCHD2* and MINOS1, which impacts both the mitochondrial structure and dynamics [31,32]. The depletion of mitochondrial DNA and impaired mitochondrial replication have been associated with early events of RV hypertrophy preceding clinical diagnosis [33]. In reperfusion injury following cardiac surgery procedures, ischemic or reperfusion injuries to the myocardium can significantly damage the mitochondrial structure and function [34,35], and thus, if the mitochondria are already damaged or dysfunctional in CHD patients prior to surgery, their outcomes may be significantly worse. Metabolic switching from glucose oxidation to glycolysis due to pre-clinical ischaemia may present an opportunity for therapeutic intervention. A reduction in the mitochondria-associated glycolysis gene PDK3 in CHD tissue suggests a measurable reduction in glycolysis, which could be measured in the future to predict functionality. Pre-surgical treatment to promote restorative and *healthy* mitochondrial function in the right ventricle, such as targeted pyruvate dehydrogenase kinase inhibitors [36] and fatty acid oxidation inhibitors [37], may promote enhanced recovery and longer-term stability of the RV post-surgery. However, further mitochondrial functionality assessments and effect on the left ventricle must be conducted on fresh tissue to target the mitochondrial dysfunction further.

In summary, the work presented here identifies the dysfunctional pathways present in RV biopsies from patients with CHD. Whilst animal models can attempt to recapitulate such aetiologies to understand the complex mechanisms at play that influence the outcomes of surgical intervention in these patients, more studies must be undertaken on primary tissue, integrating novel ways to utilise pre-existing -omic libraries. Due to the fragile nature of such patients, further non-invasive studies must be conducted to broaden the understanding of the complex mechanisms in paediatric CHD. 

## 4. Materials and Methods

### 4.1. Study Population

Twenty-four RV samples were collected from discarded myocardial tissue from patients undergoing cardiac surgery at Bristol Royal Hospital for Children, following informed consent or from control autopsy subjects (CHD RNAseq: 14; CHD histology: 5; and control histology: 5). All samples were taken from patients between the age of 0 and 16 years old. Of the CHD samples, all were diagnosed with a cyanotic pathology shortly after birth. Eleven Tetralogy of Fallot (ToF), two Truncus Arteriosus and one critical pulmonary stenosis (major aortopulmonary collateral arteries [MAPCAs]) (Appendix A) cardiac tissues were obtained intraoperatively and submerged immediately in liquid nitrogen, stored at −80 °C and processed for downstream use. All patients underwent cardiopulmonary bypass. Wax-embedded control tissue was obtained from clinical pathology. This study was approved by the ethics committee (NHS REC 19/SW/0113) in accordance with the Declaration of Helsinki. Clinical data, pre- and post-operative echocardiography and clinical diagnosis were reviewed for all patients retrospectively to confirm the diagnosis. Tissue from patients with suspected or diagnosed genetic pathologies was not used in the study. 

### 4.2. Tissue Processing and Histological Analysis

Fresh biopsies (10 mg net weight; n = 5 CHD tissue, and n = 5 control tissue) were collected from surgery from the apex of the RV immediately after the institution of cardiac pulmonary bypass and snap-frozen upon harvesting or processed immediately for wax embedding (Figure 1A). Briefly, RV tissue was embedded in paraffin, and 6 μm sections were cut using a HistoCore BioCut microtome (Leica, Wetzlar, Germany), and the slides were dried in a heated chamber. The sections were deparaffinised in xylene and decreasing ethanol concentrations, as previously described [38]. The slides were then washed in PBS for 5 min before incubating in heated sodium citrate for 30 min and blocking in 0.3% H_2_O_2_ in PBS for 20 min. The slides were then washed in tap water and placed in PBS for 5 min. Next, the slides were blocked in 4% bovine serum albumin (Sigma, UK) at room temperature for 1h in a humidified chamber. Following blocking, the slides were incubated with primary antibodies (Appendix A) and diluted in PBS with 2% BSA at room temperature for 1 h in a humidified chamber. The slides were then washed three times in PBS for 5 min before diluted secondary antibodies (Appendix A) were added in PBS with 2% BSA at room temperature for 1 h in a humidified chamber. The slides were then washed three times and incubated with streptavidin-HRP (Sigma, UK) at room temperature for 30 min before washing a further three times in PBS. Finally, the slides were developed in DAB (Vector Laboratories, London, UK) and washed in tap water. The slides were then counterstained in Gill haematoxylin for 30 s before washing twice in tap water, placed in ammonium water for 15 s and then washed in water twice again. The slides were dehydrated as before and mounted using a DPX-based media and imaged on a Slideview VS200 (Olympus, Miami, FL, USA). 

Haematoxylin and eosin staining was conducted on the deparaffinised sections. Briefly, the slides were washed in distilled water, incubated in haematoxylin for 30 s, and rinsed in running tap water and Scott’s tap water before incubating in eosin for 2 min. The slides were washed in ammonium water for 30 s before dehydrating and mounting, as before, and imaged on the Slideview VS200 (Olympus, USA). 

Picro-Sirius Red was used to determine collagen fibres under polarised light. Briefly, the deparaffinised slides were washed in distilled water and incubated in Picro-Sirius Red (Abcam, Waltham, MA, USA) for 60 min. The slides were then rinsed in acetic acid solution (0.5%) until excess staining was removed. Finally, the slides were dehydrated and cleared before mounting with a DPX media, as before. The slides were imaged by polarised light and brightfield light microscopy and merged on the Slideview VS200 (Olympus, USA). 

### 4.3. Transcriptomic Analysis

Strand-specific bulk RNA sequencing was performed on whole-tissue snap-frozen RV biopsies via Genewiz (Azenta). Briefly, total RNA from RV was isolated from 10mg of tissue with poly-A selection. Of the 24 samples submitted, 14 passed quality control and were taken forward for sequencing. Sequencing was conducted on an Illumina NovaSeq, yielding 2x150bp paired-end reads and 30 million read pairs per sample, with >80% of bases with Q30 or higher. Control samples were downloaded through the SRA toolkit, series GSE217772, containing raw RNA-sequencing FASTQ files of the RV samples of unaffected paediatric patients (1–6 years old), who had undergone traumatic brain injury but maintained cardiac function. RVs were isolated shortly after death and snap-frozen before processing and sequencing at the Children’s Hospital of Chongqing Medical University. The whole transcriptome was sequenced on an Illumina NovaSeq 6000 with paired-end reads produced. FastQC (Babraham Bioinformatics, V0.12) was used to provide quality control on the raw counts. Paired-end data were trimmed via SLIDINGWINDOW:4:20 command with Trimmomatic-0.35 [39]. Quantification was conducted using Salmon [40] (V1.10.2) via the mapping-based mode *salmon-quant* indexed against the human genome assembly GRCh38. 

### 4.4. Differential Enrichment and Network Analysis

DESeq2 (v1.40.2, R) [11] was used to perform differential expression analysis. Briefly, tximportData was used to import salmon output files, stratified by the presence of CHD (CHD, 14; ctrl, 4), and the results produced with the *apeglm* log2 fold change shrank. A principal component analysis was conducted with the *PlotPCA* function on the normalised transformed data through the *normTransform* function, and variance stabilising transformation was used to log the transform data. Gene clustering was performed with the *genefilter* package, with the top variable genes stratified by the presence of CHD through the transformed values of the matrix of normalised values, with volcano plots produced by *EnhancedVolcano*. Gene set enrichment was performed via the *fgsea* R package, utilising the KEGG, WikiPathways, Hallmark and Reactome databases and plotted via *ggplot2*. A network analysis was conducted in Cytoscape. The ClueGO Cytoscape plugin was used to visualise the non-redundant biological terms for large clusters based on the databases above on the significantly upregulated genes in the CHD population. The network was based on the kappa statistics and reflects the relationship between the terms based on the similarity of their functional genes. 

### 4.5. Transcriptomic Deconvolution

Bulk transcriptomic data were deconvoluted via *granulator* using the single-cell matrix from the healthy RV dataset GSE145154 [8]. Raw data were downloaded through the SRA toolkit, and Seurat object produced as previously described [41]. The Seurat object data were normalised with *scTRANSFORM*, and the cell types were identified, as dictated by the published data. Deconvolution was modelled via seven methods: *dtangle, nnls, ols, qprog, qprogwc, rls* and *svr. The* Pearson correlation of cell-type proportions across all methods was assessed, and *dtangle* was taken forward. The estimated cell-type proportions were normalised to 100% and presented per donor. 

### 4.6. Statistical Analysis

The Student *t*-test was used for 2-group comparisons of the normally distributed data. ANOVA was used for >2 group comparisons performed on the continuous, normally distributed data. Bonferroni correction for the multiple comparison test was applied where required. 

## Figures and Tables

**Figure 1 ijms-25-07487-f001:**
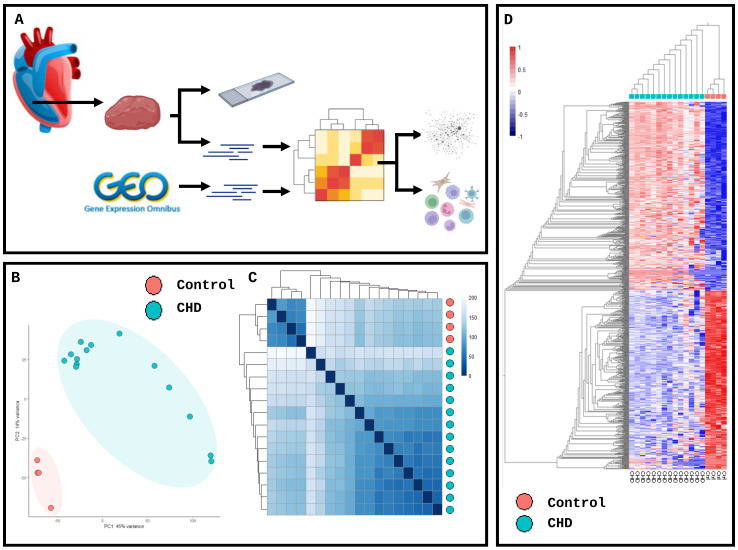
Transcriptomic analysis demonstrates a reduction in homeostatic genes and an increase in immune-infiltration regulators in congenital heart disease and control patients. (**A**) Schematic workflow of sample processing and analysis. (**B**) Principal component analysis of transcriptomic data from n = 14 whole tissue right ventricles and n = 4 control tissues from paediatric patients; no cut-off applied. (**C**) Heatmap clustering of samples based on distance matrix with hierarchical clustering applied on normalised log-transformed counts. (**D**) Hierarchical clustering of log-transformed transcriptomic profiles with a cut-off of padj ≥ 0.05 and logFC ≤ 2.

**Figure 2 ijms-25-07487-f002:**
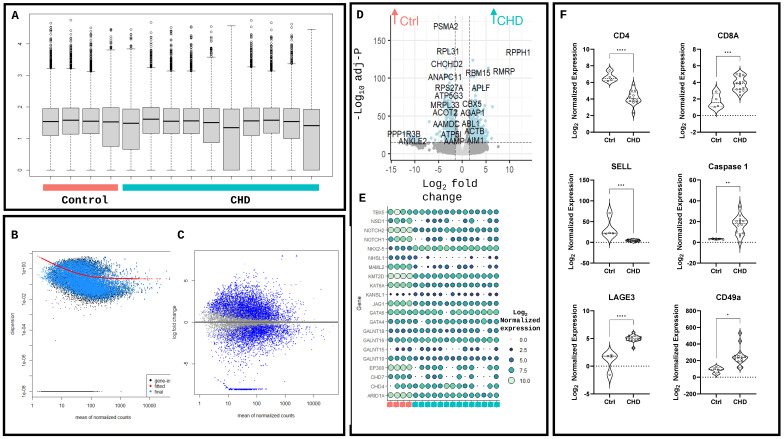
Differential expression analysis identifies hallmark inflammatory cell gene expression. (**A**) DESeq2 library normalisation expression. (**B**) Dispersion plot of normalised data. (**C**) MA plot. (**D**) Volcano plot with differentially expressed genes in control and CHD transcriptomics. (**E**) Dot plot examining hallmark CHD genes through log-normalised gene expression between control and CHD. (**F**) Immune cell genes CD4, CD8A, T-cell maturation marker SELL, inflammatory markers caspase-1, LAGE3 and T-cell homing marker CD49a. Statistical analyses were performed using paired-*T* tests corrected for multiple comparisons (n = 18). * *p* < 0.05, ** *p* < 0.01, and *** *p* < 0.001. **** *p* < 0.0001.

**Figure 3 ijms-25-07487-f003:**
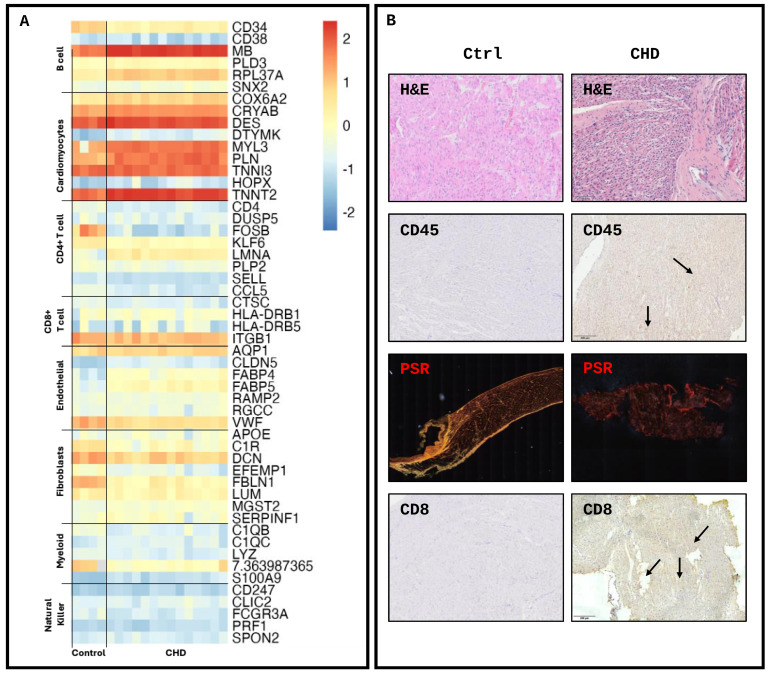
Transcriptomic and histological identification of cellular changes in right ventricle biopsy. (**A**) Heatmap of key transcriptomic markers of prominent cell types in the ventricle identified through compiled single-cell data. (**B**) Histological analysis: haematoxylin and eosin, Picro-Sirus red, and CD45 and CD8 of control vs. CHD RV biopsies. (Transcriptomics, n = 18; histology, n = 10). Scale bar = 200 μm.

**Figure 4 ijms-25-07487-f004:**
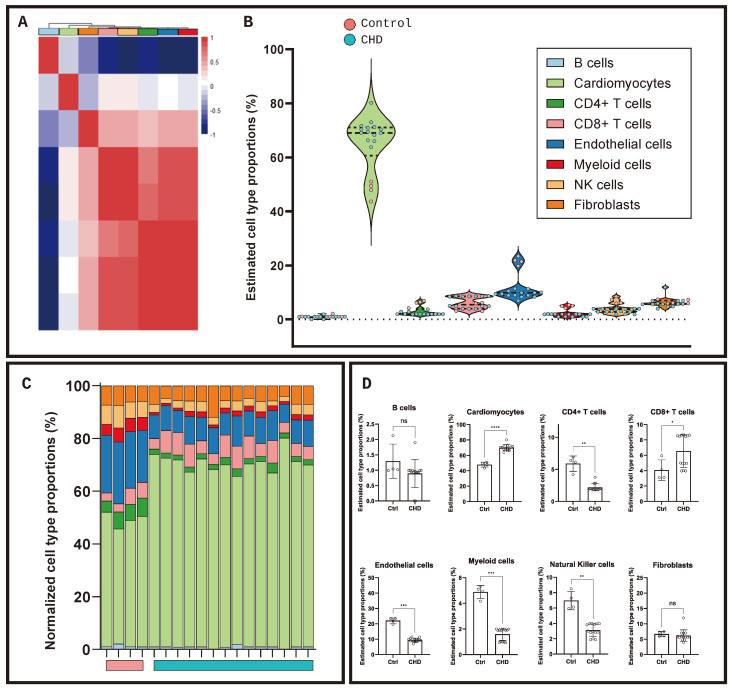
Deconvolution of transcriptomic profiles identifies eight cell types in ventricle tissue. (**A**) Correlation heatmap of cellular phenotypes deconvoluted from RNAseq data using healthy and inflamed right ventricle single-cell RNAseq datasets in human heart failure. (**B**) Estimated cell-type proportions from deconvoluted RNAseq datasets based on log-normalised TPM data. (**C**) Estimated cell-type proportion for each donor normalised to 100%. (**D**) Cell types identified in deconvolution. * *p* < 0.05, ** *p* < 0.01, *** *p* < 0.001 and **** *p* < 0.0001.

**Figure 5 ijms-25-07487-f005:**
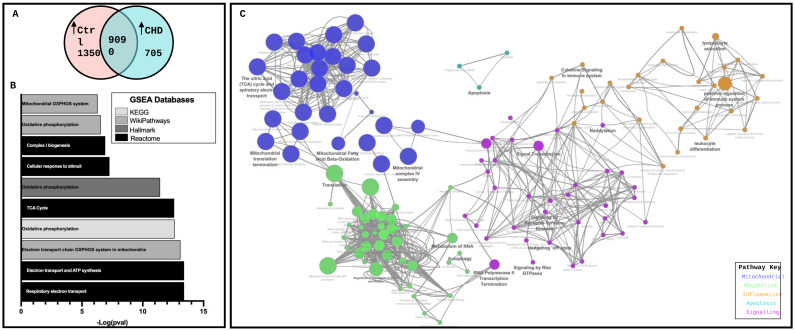
Network analysis of gene set enrichment identifies a dysregulation in inflammatory and mitochondrial pathways. (**A**) Whole transcriptomic analysis following alignment: 82% of genes identified overlap. (**B**) Top 15 significant (*p* < 0.05) gene set enrichment pathways in CHD identified from 5 databases: Human Phenotype Ontology, KEGG, WikiPathways, Hallmark gene set and Reactome. (**C**) Network visualisation of genes increased in CHD vs. control tissue. Node size corresponds to inverse Bonferroni corrected pval through the step down method; largest size; smallest pval (*p* = 0.001) smallest size; larged pval (*p* = 0.1).

## Data Availability

Raw and processed transcriptomic data are available through GEO (GSE256516). All scripts used in this study can be made available upon reasonable request to the authors.

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
