# Peer review of "Altered Inflammatory State and Mitochondrial Function Identified by Transcriptomics in Paediatric Congenital Heart Patients Prior to Surgical Repair"

_ijms, 2024, doi:10.3390/ijms25137487_

Round 1

Reviewer 1 Report

Comments and Suggestions for Authors

The authors presented an interesting manuscript assessing inflammatory activity in pediatric patients undergoing surgery due to a congenital defect. Interesting study, well presented results, well discussed.
Basically, I have no negative comments about the work. From my point of view, I would prefer that the patient population was presented to a basic extent in the "Patients and Methods" chapter, and not only in the appendix.
I believe that such an amendment is worth considering

Author Response

Reviewer 1

The authors presented an interesting manuscript assessing inflammatory activity in pediatric patients undergoing surgery due to a congenital defect. Interesting study, well presented results, well discussed.

Basically, I have no negative comments about the work. From my point of view, I would prefer that the patient population was presented to a basic extent in the "Patients and Methods" chapter, and not only in the appendix. I believe that such an amendment is worth considering.

We thank the reviewer for this kind review. As suggested, we have included the basic patient population demographics into the 2.1 study population section; the table with the demographics has also been cited in this section.

All samples were taken from patients between the age of 0 and 16 years old. Of the CHD samples, all were diagnosed with a cyanotic pathology shortly after birth; 11 Tetralogy of Fallot (ToF), two Truncus Arteriosus and one critical pulmonary stenosis (Major aortopulmonary collateral arteries [MAPCAs]) (supp. table 1). Page 1, line 59.

Reviewer 2 Report

Comments and Suggestions for Authors

This is a very interesting study regarding  the intimate molecular changes associated with congenital heart disease. The article is worth being published. I have some minor comments that should be addressed:

- it is unclear the number of subjects and controls.

- figure 3 - I recommend to change the IHC images with some with a higher magnification, to better evaluate the staining (40x).

- the introduction and discussion seem to clinical when compared to the actual results. The authors should try to better integrate the results in the case discussion, and especially in relation to other studies regarding mitochondrial disfunction in cardiac disease.

Author Response

Reviewer 2

This is a very interesting study regarding  the intimate molecular changes associated with congenital heart disease. The article is worth being published. I have some minor comments that should be addressed:

We thank the reviewer for this kind comment, we have addressed the individual points below.

  1. It is unclear the number of subjects and controls.

We apologise for this confusion. The text within the methods; section 2.1 study population; we have clarified the text.

Twentyfour RV samples were collected from discarded myocardial tissue from patients undergoing cardiac surgery at Bristol Royal Hospital for Children, following informed consent or from control autopsy subjects (CHD RNAseq; 14, CHD histology; 5, control histology; 5). Page 1, line 58.

Within the method section 2.2; tissue processing and histological analysis the text has been amended to better describe the patient numbers

Fresh biopsies (10 mg net weight, n=5 CHD and n=5 control tissue) were collected from surgery. Page 1, line 72.

  1. Figure 3 I recommend to change the IHC images with some with a higher magnification, to better evaluate the staining (40x).

We thank the reviewer for this comment. Unfortunately our lab within the hospital is currently undergoing a large scale refurbishment and thus our microscopes have been put into storage alongside all our histological samples. We currently don’t have access to these until Dec. 2024 when the building work is set to be complete and thus cannot reimage the images at the higher magnification suggested. We sincerely apologise for this oversight and within future studies we plan to image at higher magnification than 20x, in line with the reviewers comments.

  1. The introduction and discussion seem to clinical when compared to the actual results. The authors should try to better integrate the results in the case discussion, and especially in relation to other studies regarding mitochondrial disfunction in cardiac disease.

We thank the reviewer for this thoughtful comment. Please see below the changes to the introduction and discussion, further linking mitochondrial disfunction in congenital heart disease to our work, noting specific pathways and gene changes.

Introduction.

Differentially expressed genes and associated pathways demonstrate the inherent differences between tissue obtained from the left and right side of the heart4,5, however there is limited knowledge in how this compares to non CHD tissue, mainly due to the difficultly in obtaining such tissue. Inflammatory pathways within the right ventricle are suggested to develop in utero38, however the functionality of these pathways prior to reparative surgery is unknown. Notably, changes within mitochondria associated gene expression and subsequent energy production through oxidative phosphorylation can severely limit RV function, leading to pulmonary hypertension and heart failure39. Page 1, line 55.

Discussion.

This data, aligned with the network analysis predicting changes in leukocyte differentiation and lymphocyte activation, suggests a shift to a pro-inflammatory profile within the RV, which is not observed in control tissue. Page 9, line 327.

Within the RV CHD analysis, increased dysfunction within the oxidative phosphorylation and the electron transport chain were identified, in relation to in-creased expression of CHCHD2 and MINOS1, which impacts both mitochondrial structure and dynamics40,41. Page 10, line 358.

A reduction in mitochondria-associated glycolysis gene PDK3 within CHD tissue suggests a measurable reduction in glycolysis, which could be measured in future to predict functionality. Page 10, line 370

Reviewer 3 Report

Comments and Suggestions for Authors

Dear editor and authors

First of all, I want to thank you for the opportunity to review this work, which I think is very well designed and executed. So:

- The introduction is concise and exact and introduces us to the topic satisfactorily. The objective of the research is clearly expressed.

- The methodology is described perfectly and is reproducible in any medium, so I think it is appropriate. In the same way I think that the statistical analysis is designed perfectly.

- The results section is expressed perfectly and it is worth highlighting the quality of the figures, which greatly clarify the results.

- The discussion and conclusion seem very appropriate to me, especially the possible clinical implications of the results obtained. In that sense, I believe that the only modification that could be made is to structure the discussion in the subsections: summary of results, comparison with previous literature, advantages and limitations of its study, consequences for future research and clinical applicability and finally the conclusions. .

- Finally, the bibliography seems very adequate to me.

Author Response

We thank the reviewer for this extremely kind review of our work, we are happy you agree with our statistical methods and we look forward to following up this manuscript with future work delving deeper into the immunoregulatory methodology within CHD. In regard to the changes within the discussion subheadings, this is not allowed within the instructions for authors, so we have not edited the subheadings. 

Reviewer 4 Report

Comments and Suggestions for Authors

This research appears to be important. However, I have some suggestions.

This study aims to identify new research pathways regarding the mechanisms associated with RV dysfunction by identifying various profiles between CHD and control groups through organizational analysis. An introduction introducing the research trends on existing research pathways is needed.

The general characteristics of the study participants should be included.

More discussion is needed in the discussion section regarding the significantly identified genes and various markers.

Author Response

Review 4

This research appears to be important. However, I have some suggestions. This study aims to identify new research pathways regarding the mechanisms associated with RV dysfunction by identifying various profiles between CHD and control groups through organizational analysis.

We thank the reviewer for this overview. We address the specific comments below.

  1. An introduction introducing the research trends on existing research pathways is needed.

We thank the reviewers for this opportunity to expand this information. There is limited research on right ventricle compared to control tissue pathways, however we have included the recent findings on inflammation in RV compared to LV within the introduction, please see below.

Differentially expressed genes and associated pathways demonstrate the inherent differences between tissue obtained from the left and right side of the heart4,5, however there is limited knowledge in how this compares to non CHD tissue, mainly due to the difficultly in obtaining such tissue. Inflammatory pathways within the right ventricle are suggested to develop in utero38, however the functionality of these pathways prior to reparative surgery is unknown. Notably, changes within mitochondria associated gene expression and subsequent energy production through oxidative phosphorylation can severely limit RV function, leading to pulmonary hypertension and heart failure39. Page 1, line 55.

  1. The general characteristics of the study participants should be included.

We thanks the review for highlighting this. We have included all available patient information into supplemental table 1. Due to the access of the RNA sequencing data from a data repository we have limited information on these patients. I have reached out to the authors of the dataset, however they declined to share any further information regarding patient demographics until their manuscript is published.

  1. More discussion is needed in the discussion section regarding the significantly identified genes and various markers.

We thank the reviewer for this point, we have now added into mentions to specific genes and pathways throughout the discussion. Please see below for the individual references.

This data, aligned with the network analysis predicting changes in leukocyte dif-ferentiation and lymphocyte activation, suggests a shift to a pro-inflammatory profile within the RV, which is not observed in control tissue. Page 9, line 327.

Within the RV CHD analysis, increased dysfunction within the oxidative phos-phorylation and the electron transport chain were identified, in relation to in-creased expression of CHCHD2 and MINOS1, which impacts both mitochondrial structure and dynamics40,41. Page 10, line 358.

A reduction in mitochondria-associated glycolysis gene PDK3 within CHD tissue suggests a measurable reduction in glycolysis, which could be measured in future to predict functionality. Page 10, line 370.

Round 2

Reviewer 4 Report

Comments and Suggestions for Authors

Thank you for your efforts in revising the manuscript.

But, the format and font of the paper do not match the journal's regulations, please check them.

Author Response

Thank you for accepting our responses. 

The formating and the font were supplied by MDPI IJMS upon formatting by the editor and thus presumably in a appropriate to journal  regulations - i've emailed the editor to confirm if this formatting is sufficient. We are happy to reformat the manuscript, just unsure how to proceed.  

This is the response we received from the editor - hopefully this is clarified now. 

Dear Dr. Bartoli-Leonard,

Thank you for your email. I hope this message finds you well. I would
like to kindly request that you upload the updated file of your
manuscript on the SUSY system at your earliest convenience.

Once you have resubmitted your revised version, rest assured that me and
our team will promptly process it and provide assistance with any
necessary adjustments to ensure compliance with the manuscript format
and layout requirements.

We are committed to supporting you throughout this process, and we look
forward to receiving your revision soon. Your cooperation in uploading
the updated file will expedite the next steps in the review process.

Should you encounter any difficulties or have questions regarding the
submission procedure, please do not hesitate to reach out to us. We are
here to assist you and facilitate a smooth progression towards the
publication of your work.

Thank you for your attention to this matter, and we appreciate your
prompt action in uploading the revised manuscript.

Kind regards,

Ms. Nisareefah Benyakart